# Development of the Help Overcoming Pain Early (HOPE) Programme Built on a Person-Centred Approach to Support School Nurses in the Care of Adolescents with Chronic Pain—A Feasibility Study

**DOI:** 10.3390/children6090095

**Published:** 2019-08-25

**Authors:** Stefan Nilsson, Ulrika Wallbing, Gösta Alfvén, Kristina Dalenius, Andreas Fors, Marie Golsäter, Per-Åke Rosvall, Helena Wigert, Mari Lundberg

**Affiliations:** 1Institute of Health and Care Sciences, and the University of Gothenburg Centre for Person-Centred Care, Sahlgrenska Academy, University of Gothenburg, 405 30 Gothenburg, Sweden; 2Department of Neurobiology, Care and Sciences and Society, Division of Physiotherapy, Karolinska Institute, 141 52 Huddinge, Sweden; 3Clintec, Karolinska Institute, 141 52 Huddinge, Sweden; 4Lerums kommun, Sektor lärande, 443 80 Lerum, Sweden; 5Närhälsan Research and Development Primary Health Care, Region Västra Götaland, SE 411 18 Gothenburg, Sweden; 6Child Health Care and Futurum, Region Jönköping County, Barnhälsovården, Regionens hus, 551 85 Jönköping, Sweden; 7CHILD Research Group, School of Health and Welfare, Jönköping University, 551 11 Jönköping, Sweden; 8Department of Applied Educational Sciences, Umeå University, 901 87 Umeå, Sweden; 9Institute of Neuroscience and Physiology, Department of Health and Rehabilitation, Sahlgrenska Academy, University of Gothenburg, 405 30 Gothenburg, Sweden

**Keywords:** adolescent, chronic pain, person-centred care, school health care, stress

## Abstract

Chronic pain and its consequences are major global health challenges, and the prevalence is increasing worldwide among adolescents. Adolescents spend most of their waking hours in school; however, there is limited research available on how school nurses can address chronic pain among adolescents in the Swedish school context. Therefore, we designed a person-centred intervention, known as Help Overcoming Pain Early (HOPE), to enable school nurses to offer adolescents strategies to manage their stress and pain. We used the Medical Research Council (MRC) framework for developing and designing this new complex intervention. For this study, we describe two of the four phases: (a) development and (b) feasibility and piloting. The final version of the HOPE programme consists of (i) an educational package for school nurses in the areas person-centred care, stress and pain education/management and gender perspective; and (ii) an intervention package for adolescents with chronic pain. The programme consists of four sessions during which adolescents with chronic pain have person-centred dialogues with a school nurse. The HOPE programme is based on the existing evidence of managing chronic pain and on the assumption that school nurses can support adolescents with chronic pain by using person-centred care.

## 1. Introduction

Chronic pain and its consequences are major global health challenges. The prevalence of pain is increasing worldwide among children and adolescents [1]. Swedish data show that among 7613 adolescents, 12.5% report headache, 6.3% report back pain and 7.1% report stomach ache [2]. Headache and stomach ache in adolescents have been associated with psychosocial variables of anxiety and depression, childhood adversity and school stress [3]. Stress is interpreted as an important factor for chronic pain in adolescents, and an increased prevalence of perceived stress has been associated with an increased prevalence of pain [4,5]. Consequently, stressors in school are associated with a higher frequency of chronic pain in adolescents [6]. Besides, there is an association between chronic pain and impaired academic achievement among school-age children [7,8]. Adolescents spend a significant part of their waking hours in school, and, therefore, it is valuable for adolescents with chronic pain to increase their self-efficacy in school situations [9]. However, there is limited research on how school nurses can address chronic pain among adolescents in the Swedish school context. There is a lack of evidence-based interventions for adolescents with chronic pain applicable to the school context. Most school staff also lack strategies to manage chronic pain problems in the school setting, and they feel inadequately educated about how to work effectively with adolescents with chronic pain [10,11,12].

The health care system in the context of schools varies among countries. Like in other Nordic countries, school nurses in Sweden often provide the first line of support and treatment for schoolchildren [13].

School nurses in Sweden are specialists and have a particular education with focus on children and adolescents’ development and health. It is part of the school nurse’s competence to be able to assess when it is needed to consult with or refer a child to colleagues in the school health team, for example a physician, psychologist, special teacher, or social worker and to other health care actors, such as primary health care or specialised health care, such as specialized paediatric pain clinics, when necessary. Cooperation between different professions and health care levels is important and should go both ways, depending on the needs of the individual child [14].

The school health services are free of charge and reach almost every schoolchild. The school nurses are situated in the school and regularly invite the children to health visits including health dialogues according to the health-monitoring programme. The Swedish Education Act [15] stresses that school nurses’ primary role involves promoting health as well as preventive interventions. The school nurse´s knowledge about lifestyle, such as sleep, diet and physical activity are important in health promotion as well as preventive interventions [14].

However, chronic pain is a common problem among adolescents visiting the school nurse’s office [8] and cannot be resolved through health dialogues according to the health monitoring programme only. This illustrates that school nurses need to have additional strategies for supporting adolescents to cope with their pain in their everyday life.

A practice of person-centred care is advocated within the modern health care system. The World Health Organization (WHO) and national authorities in different countries highlight the importance of person-centred care [16,17]. Person-centred care strives to find strategies that increase people’s well-being and can be adapted to different ages, conditions and settings. The core feature of person-centred care consists of acknowledging the patient (or in this case, the student) as a person who actively participates in his/her health [18,19]. The WHO, however, has pointed out that research studies have so far focused mainly on treatment and diagnosis in adult and elderly care. This means that besides the evidence for different life stages (e.g., childhood), the evidence base for services such as health promotion, prevention, long-term care, treatment and diagnoses is not very strong [20].

Therefore, we designed a person-centred intervention, known as HOPE, to enable school nurses to offer adolescents strategies to manage their stress and pain. Since the research intervention was a sophisticated design, we drew on the MRC framework [21] in the development of our planned randomised controlled trial (RCT) [21,22]. MRC aims to support researchers to identify and adopt appropriate methods in their evaluation of complex interventions [21].

### Purpose

The overall purpose of this article was to describe the development and implementation of HOPE—a person-centred programme to be used by school nurses in order to support adolescents with chronic pain to increase their self-efficacy for better health. The specific objectives of this article included:developing a pain education/management programme based on an evidence-based person-centred approach and existing evidence-based stress and pain management guidelines anddetermining the feasibility of conducting an effectiveness trial of the primary outcome in terms of self-efficacy.

## 2. Materials and Methods

We used the MRC framework for developing and designing this new complex intervention [21]. The process of developing a complex intervention consists of several phases. For this study, we describe two of these phases: (a) development and (b) feasibility and piloting. The RCT is part of the third phase, ‘Evaluation’, and will be published elsewhere. The phases should not, however, be seen as linear but as part of an interactive process (Figure 1).

### 2.1. MRC Phase 1: Development

#### 2.1.1. Identifying the Evidence-Based Practice

As a first step in developing this programme, we investigated the evidence-based practice for pain management in the context of school health care. We intended to conduct a systematic literature review; however, a preliminary search revealed the shortage of articles in the school context. Instead, we used a participatory design with adolescents and school nurses.

Individual interviews are mainly used in participatory design with adolescents who can communicate well verbally to obtain views, thoughts and stories. Interviews should take place with adolescents who are interested in talking about a particular topic, and we chose to interview adolescents with chronic pain [23].

In 2014, we performed two pilot studies to gain insights into the problem of chronic pain [24,25]. One of the pilot studies was an interview study asking school nurses about how they managed adolescents with chronic pain. The authors contacted a municipality in western Sweden, and three focus group interviews were conducted. Ten female school nurses aged between 33 and 61 years participated. The school nurses had been active in school health care between 1 and 24 years. Five of the school nurses were district nurses; one of the school nurses had attended specialist training with a focus on school nursing, and four of the school nurses had undergone other specialist training. Each focus group interview lasted between 45 and 60 min [24].

The other pilot study was also an interview study and explored how adolescents managed chronic pain. The authors contacted a municipality in western Sweden, and nine individual interviews were conducted. Seven adolescents were attending occupational preparatory programmes, and two were pursuing higher education studies. The interviewed students comprised two boys and seven girls aged between 17 and 18 years. Each interview lasted between 20 and 60 min. The adolescents were invited if they fitted the definition offered by the International Association for the Study of Pain (IASP) and had reported recurrent pain during the past three months [25].

The results of these pilot studies pointed out the following areas of improvement. First, the responsibilities for adolescents with pain were not very clear in the school, and the school nurses in the study felt confused about how they should handle the various situations that arose. Second, the adolescents emphasised that school nurses should be genuinely interested in listening to their stories [25]. In 2015, these pilot studies were followed up with additional interviews, which aimed to encourage the participation of school nurses and adolescents from a different age group, school level and region in Sweden. For these interviews, 15 adolescents aged between 14 and 19 years and 15 school nurses in 10 secondary schools and 3 upper-secondary schools were recruited. These interview studies aimed to ensure that the research question achieved saturation. The results from the previous pilot studies reached saturation after these interviews, and the lack of guidelines about how to manage chronic pain led to an imbalance of available resources among the adolescents. The importance of trust was highlighted, and most of the adolescents preferred meeting a school nurse to a psychologist. They thought that it was more ‘normal’ to visit a school nurse than a psychologist mainly because their peers tended to associate less stigma with the former [26,27].

In summary, we concluded that the school nurses in the study felt confused about how they should deal with children suffering from chronic pain. The adolescents (across age groups and regions) stressed the importance of the school nurse being someone genuinely interested in listening to adolescents’ stories, and they preferred meeting a school nurse to meeting a psychologist. Drawing on these results, we designed a pain management programme applicable to the school context (Figure 2).

An expert group was formed in 2016 to build the HOPE programme. The expert group consisted of experts in person-centred care (Andreas Fors, RN, Ph.D.), stress and pain management (Gösta Alfvén, Paediatrician, Associate Professor; Stefan Nilsson, RN, Associate Professor; Ulrika Wallbing, PT, Spec Mental Health, Child and adolescent psychotherapist, MSN), gender perspective, school education (Per-Åke Rosvall, Ph.D., Associate Professor) and school nursing (Kristina Dalenius, RN, School Nurse, BSc; Marie Golsäter, RN, Ph.D.; Helena Wigert, RN, Associate Professor).

#### 2.1.2. Identifying and Developing the Theory

Given the lack of evidence-based practice for school nurses for pain management interventions in the school context, the HOPE programme was based on stress and pain education/management for children with chronic pain [28]. The overarching main theoretical framework revolved around a person-centred ethic, which characterised all parts of the programme [29], especially stress and pain education/management and the gender perspective.

##### HOPE–Person-Centred Care

According to the Gothenburg University Centre for Person-Centred Care (GPCC), person-centred care has three core components: (a) listening to the narrative, (b) establishing a partnership and (c) sharing documentation [29]. An adolescent’s narrative is important in creating a partnership by identifying his/her experiences, present situation, needs, capabilities and resources. The second step involves co-creating a health plan in line with the identified resources and barriers combined with medical and health research evidence. The third step includes documenting and following up the health plan. If necessary, the health plan is revised when evaluating the set goals [19].

##### HOPE–Stress and Pain Education/Management

In 2016, at the time of the start of the study, pain education was part of the clinical practice for adolescents who visited a pain clinic. Given the close comorbidity with stress, such an education also contained features of stress management. The education was based on the assumption that chronic pain can be defined as a biopsychosocial phenomenon [30]. The education demonstrated that stress could influence chronic pain, i.e., that adolescents with chronic pain have shown increased muscular resting activity and a potentiation of the startle reaction [31]. The education also demonstrated that environmental factors could influence adolescents´ chronic pain. Stressors in the school environment, e.g., harassment by peers, being treated badly by teachers and schoolwork pressure, could be associated with chronic pain [4]. The education showed how central sensitization could contribute to inflammatory pain, migraine, and irritable bowel syndrome. Pain is not a simple reflection of peripheral inputs or pathology but a dynamic reflection of central neuronal plasticity, which can contribute to clinical pain syndromes [32]. The content of the stress and pain education has afterwards been validated [28].

##### HOPE–Gender Perspectives

Stress-related chronic pain is more commonly reported among females than males [33,34,35]. However, this should not be understood as a genetically based difference. Even though numerous studies refer to gender-specific differences in recurrent pain [36,37], they are insufficient to fully elucidate the underlying mechanisms involved [38], such as gender-based power structures [39]. Instead, gender-based differences might be what one should expect when considering structural differences whereby women tend to be responsible for both labour work and domestic work and have differential wages and statuses in the labour market compared to men [40]. Gendered norms of how to act seems to include how to respond to recurrent pain already at an early age. Thus, studies indicate that chronic pain is more strongly related to anxiety and depression in girls than boys [41,42] whereas disruptive behaviour disorders [41] and severe suicidality [43] are more frequently associated with reports of chronic pain in boys. However, it has been reported that girls and boys experience similar levels of pain intensity when affected of recurrent pain [44]. Nevertheless, school-based interventions directed at women and mental health issues have been criticised, mainly because they tend to fail to recognise male students with mental health problems [45]. Besides, the interventions miss out on the possibility of informing both boys and girls and treating mental health issues as a structural problem rather than an individual one, such as the existence of traditional gender-based stereotypes and norms for how a female and male should act.

Since boys and girls sometimes respond differently to recurrent pain [41,43,44] and it is recommended to address recurrent pain early in order to avoid problems in adulthood [46] the introductory educational part of the intervention directed at the school nurses includes a lecture on gender awareness.

#### 2.1.3. The HOPE Intervention

Based on the theoretical reasoning above, the HOPE Intervention was developed in two steps. The first step was to develop an education programme for school nurses. The education programme was based on person-centred care [19,29] and stress and pain education/management [28]. One part in the educational programme was to strengthen the nurses’ knowledge about chronic pain conditions among adolescents and a physician with long experiences of working with adolescents with chronic pain took part in the education programme. Another important field was the gender perspective [47]. The second step was to develop an intervention for adolescents.

To co-create a health plan consistent with the identified resources and barriers combined with medical and health research evidence, we developed a health plan that consisted of four sessions (Figure 3). The content of the health plan had not been determined beforehand. The adolescent chose a stress and/or pain management strategy that was suitable for her/him (Table 1). However, based on the school nurses’ time resources, the length of the four sessions was predetermined. The health plan was inspired by an earlier version used for adults [48]. The health plan was adapted to be usable in the HOPE programme. The health plan consisted of documentation from four health dialogues with the following headings: (a) the main goal: what and how the adolescent should do to achieve the set goals and (b) the adolescent´s own resources: the adolescent’s need for support and follow-up on the main goal. The school nurse and the adolescent could also mention other health care contacts if needed, such as physicians and psychologists. The HOPE programme began by listening to the adolescents’ narratives to initiate partnership through identifying the adolescents’ experiences, present situation, needs, capabilities and resources [19]. In the first session, the adolescent and the school nurse also documented the goal for the four meetings. If the school nurse during the session becomes aware of medical factors that may affect the health of the adolescent, she/he consult the school physician or the health care centre for further treatment. The goal was followed up during the second session, which also contained stress and pain education material for adolescents in school health care. We developed a poster with information about stress and pain. Moreover, we used two films that were produced at a pain clinic in Germany. The films have recently been evaluated [49]. The selected stress and pain management strategies have shown to be effective in reducing stress and pain arising from breathing [50], massage/touch [51], physical activities [52] and relaxation [53].

The following two sessions focused on evaluating the goal and, if necessary, on rewriting the goal. During the fourth session, the adolescent and the school nurse wrote an evaluation. The documentation in the health plan was shared between the school nurse and the adolescent, and it was possible to make changes in the health plan during the programme [19].

#### 2.1.4. Modelling Process and Outcomes

The outcome measurements in the HOPE programme were selected because they reflected the core aspects of person-centred care and stress and pain education/management. No outcome specifically evaluated the gender perspective. The PedIMMPACT recommendation inspired the choice of the assessment tools. PedIMMPACT recommends assessing pain intensity, symptoms and adverse events, emotional functioning and sleep [54]; we also added self-efficacy [48]. Below, we will describe the selected instruments used to assess these areas.

##### Person-Centred Care

Self-efficacy has been used as an outcome measure in other studies evaluating person-centred care [48,55,56]. A validated outcome measurement for self-efficacy in adolescents is Holm et al.’s Self-Efficacy for Daily Activities (SEDA) scale [57], which we decided to use. The SEDA scale contains three subscales and a total of 16 items that are rated on an 11-scale Numerical Rating Scale (NRS), which are added up to give a total score ranging from 0 to 160, where higher scores indicate greater self-efficacy. SEDA is the primary outcome in this programme, and a power calculation will be conducted on SEDA when the RCT is performed. Another important outcome measure in person-centred care is self-rated health whose outcome measurement was measured in accordance with Duberg et al.’s [58] study.

##### Stress and Pain Management

Depressive symptoms are measured using the child’s version of the Center for Epidemiological Studies Depression Scale for Children (CES-DC) [59]. Pain duration, intensity, location, impact and school attendance are measured using Holm et al.’s [57] study. A questionnaire was developed and tested in a pilot study [27] (Table 2).

Sleep quality is associated with stress and pain, and the outcome measurement of the Minimal Insomnia Symptom Scale (MISS) is measured after Broman et al.’s [60] study (Table 3).

### 2.2. MRC Phase 2: Development: Feasibility and Piloting

#### 2.2.1. Ethical Considerations

Ethical approval was obtained from the Regional Research Ethics Committee in Gothenburg, Sweden (ref 172–16). The project is registered in the Clinical Trials.gov Identifier as NCT02944786.

#### 2.2.2. Testing the Feasibility of the HOPE Programme

##### Purpose of Testing the Feasibility

This test aimed to determine whether the HOPE programme was feasible and could be implemented in clinical practice according to school nurses’ views. Information about a forthcoming research project concerning how to support adolescents with chronic pain problems in school health care was given to school nurses in south-western Sweden. The school nurses received written and oral information in appointed meetings in different municipalities.

##### Testing a Pilot Version of the HOPE Programme

We used participatory co-design methods to engage school nurses in focus groups. During spring 2016, two focus groups were conducted. Six female school nurses participated in the first group, five from five secondary schools and one from an upper-secondary school; two participants were from private schools, and four were from public schools. All of them had at least one year of experience in school nursing. Five other female school nurses participated in the second focus group from three secondary schools; two school nurses were from private schools and three from public schools. All of them had at least one year of experience in school nursing. The initial schedule of the HOPE intervention was presented to the school nurses to allow for appropriate cultural adaptation to the intervention. Each focus group lasted for 60 min. The focus groups were digitally recorded and transcribed verbatim.

### 2.3. Data Analysis

After Hsieh and Shannon [61], the analysis consisted of applying directed content analysis. The main goal of a directed approach is to validate or extend a theoretical framework. This analysis applied the three core components of the HOPE programme, namely person-centred care, stress and pain education/management and gender perspectives.

## 3. Results

The focus groups investigated whether the components of the HOPE intervention had increased the school nurses’ awareness of the value of those concepts and perspectives for the school health care. The results showed that the construction of the HOPE intervention was consistent with the results from the previous interviews.

However, the interviews showed that as the school nurses increased their awareness of the intervention, they showed greater tendency to gain more knowledge of the concepts beyond what the HOPE intervention offered, for example about mindfulness and how to adapt gender perspectives.

### 3.1. Person-Centred Care

The school nurses had not previously worked according to person-centred care. They described the value of starting with the narrative. Even if the school nurses had earlier listened to the adolescents, the person-centred approach increased their motivation for listening to the adolescents’ views. The results illustrate the value of person-centred dialogues in school nursing, although this method introduces a new way of beginning by listening to the adolescents’ stories. Previously, the school nurses had used motivational interviewing; however, it had hardly culminated in a partnership whereby the school nurse and the adolescent jointly identified a goal based on the adolescent’s experiences, present situation, needs, capabilities and resources. A person-centred approach facilitated shared decision-making in the health plan, and the school nurses thought that it was important to implement person-centred care in their work: ‘I also think that it should be person-centred, starting from the person’s narratives’ (school nurse 6, focus group interview 1).

The school nurses thought that it was important to adopt a person-centred approach and genuinely listen to the adolescents’ stories. The main barrier, however, was the lack of time. The health plan in the HOPE intervention gave the school nurses a structure and model of how to work according to the person-centred approach. The adolescents normally take a lot of the school nurses’ time, and the school nurses appreciated having a structure for their meetings.

### 3.2. Stress and Pain Education/Management

The use of stress and pain education/management is valuable for school nurses, and they thought that the content of stress and pain education/management in the HOPE intervention increased their knowledge. Although some of the education materials were repetitive, the school nurses believed that the education materials enabled them to meet the challenges of stress and pain in adolescents. The school nurses highlighted the importance of stress and pain education and management, and they complained about the lack of strategies to support adolescents with chronic pain. One school nurse commented thus: ‘I would like to get some knowledge about mindfulness that I could share’ (school nurse 3, focus group interview 1).

The school nurses barely had any previous education in stress and pain management, and they thought that education was necessary to support adolescents with chronic pain. Crucially, the school nurses felt comfortable about teaching adolescents about stress and pain management strategies. Otherwise, school nurses would be unable to offer useful advice to adolescents with chronic pain.

### 3.3. Gender Perspectives

The results showed that the HOPE intervention increased the school nurses’ awareness of how gender perspectives influenced their attitudes and values when meeting the adolescents. The HOPE intervention gave the school nurses a structure to align their attitudes with gender perspectives. The school nurses highlighted the importance of gender perspectives: ‘Gender perspective since women’s health is worse. How to attack the gender differences? Would like to do something there’ (school nurse 3, focus group interview 2).

This excerpt implies that school nurses are keen to increase their gender awareness. However, it is important to continue supporting school nurses to address gender inequalities. The excerpt does not help us understand how school nurses can address the fact that ‘women’s health is worse’. Thus, school nurses run the risk of reproducing the failure of some earlier interventions in that identifying one gender inequality (e.g., the commonly reported mental health problems among women) can lead to creating another (leaving one gender out of the intervention). Thus, the focus groups appreciated the value of the gender component in the HOPE intervention in school health care. Nonetheless, for greater gender awareness, they may need further guidance.

### 3.4. The Final Version of the HOPE Programme after Cultural Adaption

The final version of the HOPE programme has only a minor cultural adaption, which is a PowerPoint presentation concerning stress and pain management that school nurses can use. The final version of the HOPE programme consists of an educational package for school nurses. This package contains seven films and seven lectures on the following topics: person-centred care, person-centred dialogues, student-centred dialogues, the structure of the conversations, gender perspectives, stress and pain physiology and stress and pain management. The main focus in the stress and pain education was to update the school nurses´ knowledge base on pain and support their conceptualization of pain from that of a marker of tissue damage or disease to that of a complex phenomenon that needs to be understood from a bio-psycho-social perspective. The school nurses became aware that credible evidence of danger to body tissue can increase pain [30]. The final version of the HOPE programme also offers an intervention package for adolescents with chronic pain. Adolescents with chronic pain have dialogues with a school nurse in four sessions. The first session concerns an adolescent’s story and needs. This story will end up in a partnership whereby the adolescent and the school nurse agree on a goal for the intervention, which is documented in a health plan. The adolescent and the school nurse jointly decide on a goal that is possible to fulfil in four sessions over six weeks. The second session takes place a week later in which the school nurse and the adolescent watch a film about pain physiology [49] and discuss the neuroscientific reasons for biopsychosocial stress and chronic pain. The school nurse can draw on posters, films and PowerPoint presentations concerning stress and pain management. The school nurse also follows up the agreed goal from the first session. The two remaining sessions are follow-up sessions, which are conducted every second week. Additionally, the adolescent’s goal is evaluated during the fourth session.

The primary outcome measure is self-efficacy (SEDA), and the secondary outcome measures are depressive symptoms (CES-DC), pain duration, intensity, location and impact as well as school attendance and self-rated health and sleep (MISS).

The HOPE programme aims to equip school nurses with strategies to manage adolescents’ chronic pain in their ordinary work and support them in their everyday life. The programme is not a replacement for physicians’ and psychologists’ formal examinations, and school nurses can refer to other professionals whenever this is necessary. School nurses are encouraged to contact other health care professionals when necessary.

## 4. Discussion

This study discussed the development of an intervention programme known as HOPE. Following MRC, this study was the first step in the process of developing and evaluating a complex intervention [21] that can help school nurses and adolescents manage chronic pain. The process included a co-design of school nurses’ and adolescents’ needs, capabilities and resources essential to the building of the HOPE programme. Moreover, health care professionals and expert researchers in the field also play a key role in the process.

In this study, person-centred care was introduced as the overall theoretical framework that can influence the development of the HOPE programme and the final version of the HOPE programme. Research has shown that adults with chronic pain prefer services with higher levels of person-centred attributes [62]; thus, the assumption is that adolescents and school nurses also prefer person-centredness. Accordingly, the HOPE programme, in all its parts, revolves around a person-centred ethic. The results of the individual interviews and the focus group interviews in this study supported these assumptions. The school nurses preferred using a model that was built on listening to a narrative, establishing a partnership and sharing documentation [29].

This is the first study that focuses on school nurses’ roles in stress and pain education/management. This type of education is traditionally given to adolescents in specialised health care. The ability to offer pain neuroscience education in the school health programme plays an important role in the building of the HOPE programme. In the HOPE programme, the school nurses adapt the education to the adolescent’s story about his/her chronic pain. The stress and pain education/management in the HOPE programme is combined with a chosen activity. The activity chosen in the HOPE programme derives from the adolescent’s narrative, and the type of activity varies depending on each adolescent’s choice. However, even if the effects of stress and pain education/management are promising, the effects need to be evaluated in an RCT.

One of the initial studies in the development of the HOPE programme reported stereotyped values arising from school nurses’ views of gender [47]. These stereotypes can hinder dialogue with adolescents, which renders an increased awareness of gender perspectives necessary. Accordingly, the HOPE programme offers education about gender perspectives in conjunction with managing adolescents with chronic pain. Having an awareness of attitudes about gender in society can facilitate school nurses’ work with a person-centred approach. Otherwise, school nurses run the risk of being influenced by stereotyped attitudes about gender. The HOPE programme contains a lecture and a film about gender perspectives for about 35 min. Besides, school nurses read an article that compares the HOPE and DISA programmes [47]. DISA has been criticised for reinforcing the stereotypical notions that girls are depressive and have low self-esteem [63]. The HOPE programme aims to minimise the risk of stereotypical interventions and embrace all adolescents with chronic pain. It is unclear whether the HOPE programme can succeed in this particular goal, and it is necessary to investigate whether the HOPE programme fulfils the ambition of inclusion and equality. The next step in developing the HOPE programme is to conduct an RCT.

### Limitations

This feasibility study has some limitations. It was not possible to conduct a member check with the school nurses who participated in the two studies that created the HOPE programme; therefore, the results are based on the interview data from other school nurses. Another limitation of the data is that we did not interview the adolescents when testing the feasibility of the HOPE programme. Instead, the interviews with adolescents were conducted in the first phase and, thus, contributed to the content of the HOPE intervention.

## 5. Conclusions

The HOPE programme is based on the existing evidence of person-centred care and the management of chronic pain; it offers school nurses and adolescents strategies to manage adolescents’ chronic pain. The programme is adapted to fit schools’ health programmes, and school nurses can implement it in their everyday work.

## Figures and Tables

**Figure 1 children-06-00095-f001:**
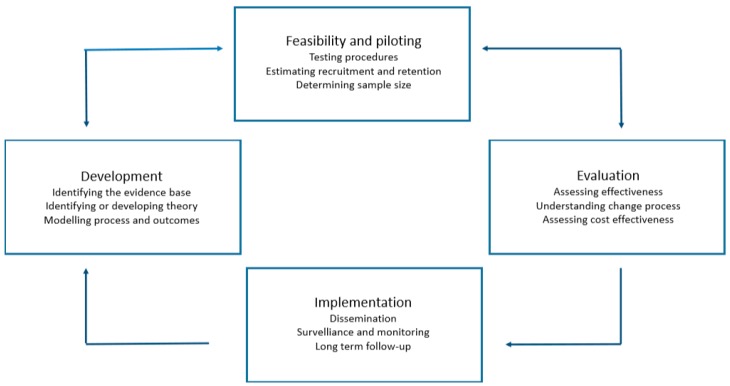
Key elements of the development and evaluation process as described by the Medical Research Council (MRC) framework for developing a complex intervention.

**Figure 2 children-06-00095-f002:**
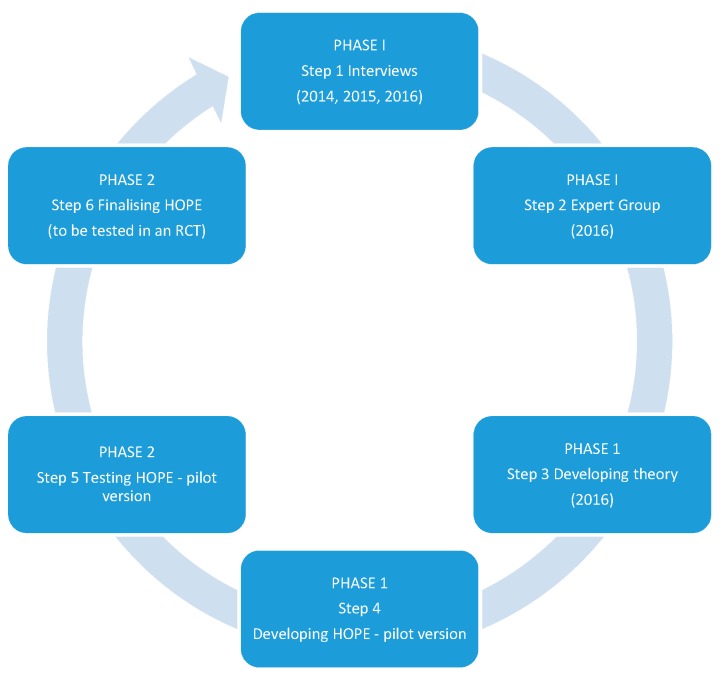
The development of the HOPE programme.

**Figure 3 children-06-00095-f003:**
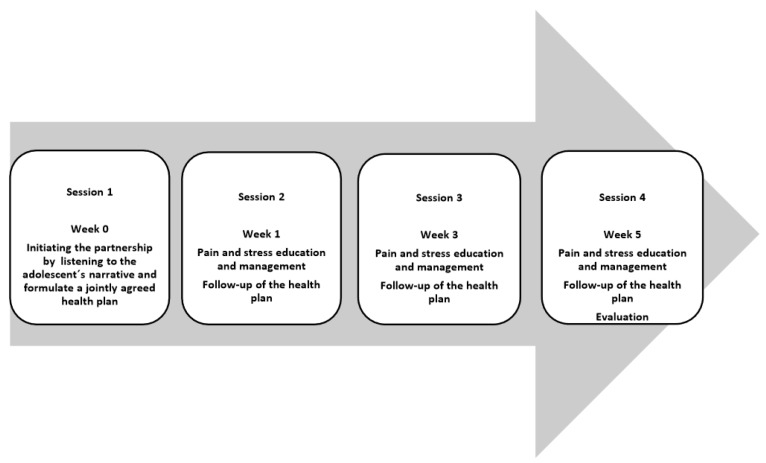
The four sessions in the Help Overcoming Pain Early (HOPE) intervention.

**Table 1 children-06-00095-t001:** Examples of stress- and/or pain-relieving strategies.

○ Breathing exercises ○ Massage/touch ○ Physical activities ○ Relaxation

**Table 2 children-06-00095-t002:** Assessments of pain location, duration, intensity and impact.

The most dominant pain localisation
The second most dominant pain localisation
The frequency of the most dominant pain
*Continuous pain*
*Several times a day*
*Once a day*
*Several hours each time*
*Several times a month*
*Once a month*
The duration of the most dominant pain
*Continuous pain*
*Several days each time*
*Several times a week*
*Approximately one hour each time*
*Several minutes each time*
The intensity of the most dominant pain (0–10)
To what extent does all the pain influence you?
*Not at all*
*Pretty much*
*Little*
*A lot*

**Table 3 children-06-00095-t003:** The constructs assessed for each of the programme’s key domains.

Construct	Measurement
Depression	CES-DC [59]
Pain duration, intensity, location and impact	Questionnaire (Table 2)
School attendance	Journal data
Self-efficacy	SEDA scale [57]
Self-rated health	Self-rated health [58]
Sleep	MISS [60]

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
