# Peer review of "Development of the Help Overcoming Pain Early (HOPE) Programme Built on a Person-Centred Approach to Support School Nurses in the Care of Adolescents with Chronic Pain—A Feasibility Study"

_children, 2019, doi:10.3390/children6090095_

Round 1

Reviewer 1 Report

The authors have identified a need for targeting adolescent chronic pain via schools and specifically via school nurses. This is an important approach that warrants careful investigation and consideration.

This approach seems useful for many albeit not all school systems. For example, government-funded schools in several counties that I am most familiar with do not have school nurses. School administrative staff provide basic first aid and manage the “sick-bay”. Nevertheless, school that do have nurses (and presumably there are many that do) may benefit from this type of approach.

I have listed some general issues as well as some specific points for the authors to consider. Although I did devote considerable attention to carefully reading the paper, I felt that there were various things that I remained somewhat confused about. The paper would benefit from greater clarity in places to help make the reader’s job easier. I have tried to identify areas that need improvement in my comments below:

General issues:

Assuming that numerous adolescents with chronic pain have not been seen by a doctor or Pain Clinic, how can the school nurse assume that the child does not in fact have some other medical condition that requires medical management (instead of encouraging the adolescent to adopt biopsychosocial strategies)?  I would think that if the school nurses are implementing a chronic pain intervention they would need confirmation from a doctor that a thorough medical examination has been carried out to rule out other conditions that need medical intervention.

If the adolescents have been seen by a doctor or Pain Clinic, do the school nurses make contact with these health professionals to ensure that all professionals are working towards the same goals?

The paper has the potential to serve a useful function of providing an overview of the HOPE program. The data provided in the paper is based on interviews carried out with 11 school nurses (6 in the first group and 5 in the second group – I assume that they were different nurses in the two groups).  It would be good if it was made clearer which statements in the Results were specifically derived from the interviews. Some statements that are made seem more like pre-conceived assumptions. For example Page 8, line 280 “The use of stress and pain education/management is valuable for school nursing…” Was this part of the sentence based on the current interview data or was it derived from something else, e.g. evidence reported elsewhere or clinical experience?

I would encourage the authors to more comprehensively review the existing body of literature about how schools can students with chronic pain. I agree with the authors that there are significant gaps in this literature. Most of the literature relevant to schools has adopted a focus on the school functioning of children with pain and not school-based interventions. For example, you may wish to consider some of the following:

Deirdre E. Logan, Laura S. Gray, Christina N. Iversen, Susan Kim, School Self-Concept in Adolescents With Chronic Pain, Journal of Pediatric Psychology, 42(8), September 2017, 892–901.

Logan, DE & Curran, JA. (2005). Adolescent chronic pain problems in the school setting: Exploring the experiences and beliefs of selected school personnel through focus group methodology. J of Adol Health, 37(4), 281-288.

Logan DE, Simons, LE, Stein, MJ & Chastain L. (2008). School impairment in adolescents with chronic pain. J of Pain, (5), 407-416.

Logan, D. E., Catanese, S. P., Coakley, R. M., & Scharff, L.(2007). Chronic pain in the classroom: Teachers’ attributions about the causes of chronic pain. Journal of School Health,77, 248–256.

Chan, E., Piira, T., & Betts, G. (2005). The school functioning of children with chronic and recurrent pain. Pediatric Pain Letter,7, 11–16.

Sato, A. F., Hainsworth, K. R., Khan, K. A., Ladwig, R. J.,Weisman, S. J., & Davies, W. H. (2007). School absenteeism in pediatric chronic pain: Identifying lessons learned from the general school absenteeism literature. Children’s HealthCare, 36, 355–372.

Specific points:

Please reference the second sentence about the increasing prevalence of pain.

I think the authors can build a stronger case for why school-based interventions can be helpful – eg perhaps citing problems that this cohort has with school attendance and functioning effectively at school if they do attend.

Page 2, line 44. I would suggest changing the word “most”  to a significant or large part of his/her waking hours.

Page 2, lines 65-69. This paragraph is a little too conversational. I would suggest avoiding expressions like “we wanted to” and “we decided to”.

Page 2, line 68 – it would help the reader if the authors explained what the Medical Research Council framework is about.

Page 3 from line 95 – “we performed two master theses” – I suggest changing the terminology. It does not seem relevant what academic level these projects were as long as they were well implemented. If they weren’t published elsewhere these may be cited as unpublished dissertations as you have done. Were these pilot studies? Or preliminary studies?

Page 5 – It is a vague/weak statement to say that “The education in HOPE was based on earlier clinical experiences and research of paediatrician and Associate Professor Gösta Alfvén.” A far stronger basis would to cite the research of Alfen or others which has helped shape the program. The subsequent sentence does provide a citation, however, more detail is needed. What interventions were chosen for inclusion, and what was the evidence in support of their decision?

Page 5 describes the importance of adopting a focus on gender issues when addressing mental health issues. While this is true, it would be helpful for the authors to review the literature more specifically on gender issues related to paediatric chronic pain, as there is a sizeable body of literature in this area as well.

Page 6 – line 193. Make it clearer in what way the assessment tools were inspired by the PedIMMPACT guidelines. Presumably by utilising a multi-dimensional approach to the assessment of outcomes.

The authors mention specific tools for assessing depression and insomnia. However, they only mention the construct of “impact of pain” without specifying how they would assess impact nor functioning. It may be more appropriate to leave all the precise details (ie all the specific measures) for how the program will be evaluated for the subsequent paper. Nevertheless it may be useful to mention the constructs that should be assessed as indicators of program efficacy. I wonder whether this may be done as a table, listing the constructs that should be assessed for each of the key program domains.

I don’t understand the heading 2.1.3 Determining Sample size. Most of the content in this section does not seem to match the heading.

More detail is required on the nature of the pain education given to nurses and to adolescents. If the adolescent and school nurse don’t share the same view about the pain, this will compromise their ability to work together effectively. How do the School nurses respond if the adolescent has a very strongly medical model of their pain?

As noted in my general comments, the manuscript does not seem to describe the nurses liaising with the adolescent’s treating health professionals. It is generally considered very valuable for the school to be in contact with the health professionals (e.g. Pain team) working with the adolescent/family. Was collaboration between nurses and health professionals encouraged?

More information is needed when the authors refer to a “health plan” (section 2.1.3). It is stated that the health plan “was inspired by an earlier used health plan for adults (28).” The reference given is for person-centred care with adults following acute coronary syndrome. Given that is a very different context, it would be useful for the authors to be clearer on what they drew from that context and what was specific to the current context.

Does the School Nurse liaise at all with the class teachers?

I found the structure of the paper a little awkward. If reporting the results of a feasibility study, I would think a Participants section would be required. The section on Page 7 Live 219 titled Aim also seemed a little out of place. I would have expected the aims to be at the end of the Introduction along with the existing Purpose section.

The Results section should present results only, and not introduce Limitations (e.g. P7, line 243). These should go in the Discussion. It is a considerable limitation (as the authors noted in the Results section) that adolescent perspectives were not elicited when assessing whether the program feasibility and acceptibility. Were there other limitations?

There is a section in the Results section titled “the final version of HOPE”. This implies that the program may have been modified in response to feedback. I had understood that the program had been developed but that the focus groups were used to make any “cultural adaptations” – what changes were made? Please clarify and consider whether the specific details of the program belong in the earlier sections of the paper or in the Results section.

In the Discussion Page 9, lines 332 and following, the authors provide information about the program’s content that seems more fitting to the Introduction, and it was not clear to me why it was included in the Discussion.

The Conclusion section does not seem to be based on the data being presented in the current paper, but rather the assumptions that the authors had at the outset of the Program. This further adds to the impression that the data presented in this paper is not the main focus of the paper, but rather the main focus is to describe the program itself.

Reviewer 2 Report

Thank you for the opportunity to review this paper.  This is one of the first papers I have seen addressing the role that school nurses could provide to help teens with chronic pain. Such an important opportunity to support these nurses to help teens manage chronic pain. Developing a structured program that school nurses could implement with teens is an important step in helping teens manage pain in their school environment.  

I did find several gaps in the paper and will address those here.

I found the statement on line 23 in the abstract and again on lines 44-46 citing the lack of research on how to address chronic pain in the school context surprising, given that there are several papers (10 that I am aware of)  addressing school functioning and chronic pain.  I think it would be helpful for the authors to address this literature.  

in the section on development, the authors discussed the participatory design.  I was unable to review the two master's theses cited and I suggest adding in a section on the two populations included in this research - how many school nurses and how many adolescents, how were they recruited, how were the students identified as having chronic pain.   

In developing the HOPE intervention for school nurses, the authors briefly touch on pain neuroscience education as a component of the intervention, but the bulk of the intervention discussed in the paper relates to stress management. I am concerned that the when  biological basis of pain is not addressed , and the focus of the intervention is on stress management only, the school nurses will not be able to address the importance of the biological basis of pain in the bio-psycho-social model. Could the authors speak more to what is included in the pain education program provided to he school nurses.

section 2.1.3 Modelling process and outcomes.  This section 

on determination of sample size. - the sample size required is not addressed and the authors state it will be determined prior to the RCT.  Given that this paper reports on the pilot version test I'm not sure the section adds to this paper. 

In the aim section, please explain how the school nurses were recruited to the focus groups. 

The results section provided an overview of the focus group findings and an important finding identified was the effect of the person- centred approach, which  was different that the previous approach of motivational interviewing that the school nurses had been using.  The nurses identified the lack of time, and how using a the structured program may help them use the limited time effectively. the time required by the nurses to complete the education package was included in the paper.  It would help to strengthen this section to have additional information about the content of the package, especially related to pain education.   It would be helpful to include the list of the 7 films and 7 lectures. 

The paper briefly touches on the intervention package for the adolescents.  additional information on what is in the package would be helpful.

In the section on gender, beginning on line 153, and continuing through the discussion, the authors cited one reference (24) on stress-related chronic pain in adults.  The rest of the section references addressed mental health globally,  the section could be strengthened by review of the current evidence on chronic pain with a gender lens, such as the King study (2011) that addressed epidemiology across gender for pediatric chronic pain conditions.   It would also be helpful to address sex related differences rather than gender related differences as sex is biologically determined. 

I look forward to seeing the results of the interventional trial.  

Round 2

Reviewer 1 Report

The authors have addressed numerous issues raised in the previous review. However, there are a number of issues that remain:

1) I agree that it would be beneficial having someone who is knowledgeable about chronic pain and able to support adolescents with chronic pain at school. However, I remain concerned that the school nurse will often need to work in the absence of a clear diagnosis. Without a comprehensive assessment, it will not be clear to them whether the adolescent has chronic pain with no current physical pathology, or whether there is a medical condition with underlying physical pathology contributing to the adolescent’s pain, which would benefit from some intervention. With such uncertainty it is difficult to educate the adolescent about chronic pain. (E.g. can they use the analogy that chronic pain is when there is an error in the individual’s software, rather than their hardware – if the “hardware” has not first been checked out.)

I am not sure that the additions on page 11, lines 396-399, regarding the school nurse making external referrals when necessary, is enough to address this issue.

I am pleased that the authors have stated in their response letter that “shared documentation between patients, school nurses and other health professionals is an important aspect of person-centred care”.

2) The authors have made a distinction that the HOPE program is first line of care, relative to the third line of care offered by Pain Clinics. It would have been useful to more clearly articulate (or perhaps even present a model) in what way health professional responses in the first line of care should differ from the 3rd line responses and why. Are the goals different? If so in what way? This is an issue which keeps popping up but has not been satisfactorily addressed in the manuscript. (This point is related to point 1.)

3) I was impressed with how quickly the authors were able to respond to the reviews. However, I feel that they perhaps did not have opportunity to explore the literature as recommended regarding gender and adolescent chronic pain.  I had been hoping to see a nuanced discussion of reasons for why gender is a relevant factor in the context of adolescent chronic pain, drawing where possible from the adolescent chronic pain literature. The reasons offered for why a consideration of gender is important seem rather broad, and it would be helpful to be more specific to the adolescent chronic pain context. For example, one paper that comes to mind is:

Kaczynski, Claar & Logan (2009). Testing gender as a moderator of associations between psychosocial variables and functional disability in children and adolescents with chronic pain, J of Ped Psychol, 34(7), 738-748.

A thorough search would no doubt reveal other papers of relevance.

4) I overlooked previously mentioning that the purpose of the HOPE program described on Page 2, lines 74-75 does not refer to pain: “a person-centred programme to be used by school nurses who try to support adolescents’ self-efficacy for better health.” Given the program is specific to the chronic pain context, this should be stated in this definition.

Reviewer 2 Report

Thank you for your responses to my comments and for the changes you have made to the paper.  the addition of the section on the two master's theses which shaped this work was very helpful, as was addressing recruitment. 

I think the title should reflect that program is "to support school nurses " needs to be incorporated ( I know it will make the title longer but the paper  addresses helping school nurses, not directly helping adolescents)  

in line 142 please include  evidence based practice "for school nurses" for pain management

References 27,  28 and 31 were added to the section on addressing stress and pain physiology however,  did find these references didn't really address  my point on understanding pain sensitization.  additionally, the King et al reference 32, which is a great paper on epidemiology of chronic pain, does not address how pain sensitization works.  Perhaps the authors could review Woolf's paper on central sensitization (JPain 2009, 152 S2-S15 ) and Latremoliere & Woolf's paper in (J Pain 2009 10 895-926 ) for more context of pain as a biological construct.   

It is still unclear to me as to what is in the education package for nurses to help them understand pain physiology and pain management (line 381).  did you use available films such as the Understanding pain in 5 minutes?  or Lorimer Mosely's ted talk?  please provide at least one example of the films inlcuded.  

Thank you for the opportunity for me to address this.  Im really trying to make this paper as excellent as it can be, so that when you publish your intervention study we can hopefully replicate this work! such an important role for school nurses (I wish we still had them in my country).

Round 3

Reviewer 1 Report

The additional pain-specific gender literature that has been added is appropriate and helpful.

 I have no further comments to add.

Reviewer 2 Report

I have no further comments to add.